# What Influences Miners’ Safety Risk Perception?

**DOI:** 10.3390/ijerph19073817

**Published:** 2022-03-23

**Authors:** Shu Zhang, Xinyu Hua, Ganghai Huang, Xiuzhi Shi, Dandan Li

**Affiliations:** 1School of Resources and Safety Engineering, Central South University, Changsha 410083, China; zhangshu@csu.edu.cn (S.Z.); 205512131@csu.edu.cn (X.H.); baopo@csu.edu.cn (X.S.); 215511052@csu.edu.cn (D.L.); 2School of Civil Engineering, Central South University, Changsha 410083, China

**Keywords:** knowledge level, organizational safety atmosphere, organizational trust, risk communication, risk perception

## Abstract

The risks faced by the mining industry have always been prominent for every walk of life in China. As the direct cause of accidents, individual unsafe behaviors are closely related to their risk perception. So, it is important to explore the factors affecting miners’ risk perception and analyze the influencing mechanisms between these factors and risk perception. The questionnaire survey method was used to collect the data of risk perception from nearly 400 respondents working in metal mines in China. Exploratory factor analysis and confirmatory factor analysis were used to analyze and process collected data. The impact of four factors affecting miners’ risk perception was verified, namely: organizational safety atmosphere, organizational trust, knowledge level, and risk communication. Then, regression analysis, Pearson correlation analysis, and structural equation model analysis were used to examine the effect of the four influencing factors on miners’ risk perception. The four influencing factors all have a positive impact on miners’ risk perception; knowledge level has the largest explained variation of miners’ risk perception, followed by risk communication. Organizational trust and organizational safety atmosphere have an indirect and positive impact on miners’ risk perception intermediated by knowledge level and risk communication. The results offer four important aspects of mine safety management to help miners establish quick and accurate risk perception, thereby reducing unsafe behaviors and avoiding accidents.

## 1. Introduction

The risk of the mining industry always ahead of various industries in China. According to statistics, from 2003 to 2021, the number of mine accidents is 5930, which is 24.49% of all accidents, and the death toll reached 18,567, which is 21.58% of the total death toll, only ranking behind road accidents [1]. So, the accidents of the mining industry are frequent and especially serious [2]. The coal operational environment is complex, volatile, and unfavorable, including poor light conditions, environmental air containing high concentrations of dust, and much crossover work, which increases the accident rate. Frequent accidents bring miners great psychological and physical harm. So, the issue of how to decrease accidents in mines is an urgent problem. As a prerequisite for behavioral decision making, perception is closely related to unsafe behaviors, which are the central causes for accidents.

Slovic considered risk perception to be the understanding and intuitive feelings of objective ventures [3]. Sitkin and Pablo thought risk perception was the assessment of risk controllability, risk probability, and confidence of estimation by describing and feeling objective risks [4,5]. In this paper, we consider risk perception to be the feelings, understanding, and direct evaluation of objective risk through experience or knowledge. Our study shows that miners’ safe behaviors and risk perception at work need to be improved [6], and few studies concentrate on miners’ risk perception. So, it is necessary to study the factors affecting miners’ risk perception and analyze the influencing mechanisms between these factors and risk perception. This study mainly solved two questions: Which factors have a significant impact on miners’ risk perception? How did the influencing factors affect miners’ risk perception? This study offers a new approach to improve miners’ risk perception and reduce unsafe behaviors, improving influencing factors of risk perception.

According to the psychometric paradigm, socio-cultural theories, social amplification of risk, and other related theories [7,8], the main influencing factors of risk perception could be divided into two aspects: individual and organizational factors [9].

### 1.1. Individual Factors

During previous exploration of risk perception in the public, some studies found that there were significant relationships between risk perception and individual factors such as age, gender, educational level, organizational trust, career satisfaction, and so on [10,11,12,13,14].

Between the public and experts, the varied COVID-19 risk perception was mainly due to different COVID-19-related knowledge levels [15]. A high level of knowledge had both a positive and negative impact on risk perception [16]. The impact of knowledge level on risk perception should be considered in specific situations.

Public trust could significantly affect public risk perception [17]. Some studies found that institutional trust may influence individual risk perception [18,19]. Therefore, organizational trust may also affect miners’ risk perception.

Previous studies showed that the influence on individual risk perception was different among different careers [17,18,19,20,21,22]. Different careers will bring different levels of satisfaction, and a previous study showed that the higher the career satisfaction, the lower the capacity for risk perception [23]. So, it is significant to explore the impact of career satisfaction on miners’ risk perception.

Riddel found that risk attitude had a significant impact on individual judgement, which is related to their risk perception [24]. Different kinds of risk attitudes had different levels of influence on individual risk perception [25]. The way that risk attitude affects miners’ risk perception is worth exploring.

### 1.2. Organizational Factors

Security matters are related to many organizational factors, such as organizational safety atmosphere and risk communication [16,26,27].

Safety atmosphere is mainly used to describe individuals’ judgement of safety value in specific environments [6]. A previous study proved that safety atmosphere profoundly influences individual risk perception [28]. Weyman concluded that safety culture, which included safety atmosphere, had a positive impact on personal risk perception [29]. In this paper, organizational safety atmosphere refers to the direct perception of safety in a mining organization’s internal environment, such as safety atmosphere in a working environment.

Risk communication is an interactive process for exchanging risk information and opinions [30]. Sufficient and effective risk communication has a positive impact on individual risk perception [31,32,33]. It is necessary to explore the relationship between risk communication and risk perception.

### 1.3. All Influencing Factors of Risk Perception

Based on the theories above, this study focused on the risk perception of metal miners, and analyzed some influencing factors of risk perception to explore the relationships between risk perception and those influencing factors. We propose that knowledge level, organizational trust, career satisfaction, risk attitude, safety atmosphere, and risk communication may affect miners’ risk perception. Considering the mining industry’s characteristics, we analyzed the mechanisms between these factors and risk perception using the regression analysis, Pearson correlation analysis, and structural equation model analysis, and established a hypothesis model of miners’ risk perception. We designed an influencing factor measurement scale and a risk perception measurement scale, then conducted surveys. The results of this study are beneficial to expand on related theories of risk perception and, specifically, improve miners’ risk perception to effectively prevent unsafe behaviors and accidents.

## 2. Materials and Methods

### 2.1. Questionnaire and Sample

#### 2.1.1. Pre-Questionnaire and Sample

Based on the psychometric paradigm, previous studies, and verified risk perception measurement scales [34,35,36], our initial scale was revised by 15 experts in company management, 15 university teachers engaging in risk perception, and 20 graduates whose majors related to mining safety. Items referred from references are as follows: (1) hiding troubles of gas which were not solved [34]; (2) I would like to take risks to finish work more quickly and easily [35]; (3) sometimes, I will do the wrong operation and ignore warning signs [35].

After that, the first survey of initial questions was conducted. The questionnaire included three parts: demographic variables, study of influencing factors on miners’ risk perception, and measurement of miners’ risk perception. Except for the basic information, all questions were measured using the 5-point Likert scale (anchored by the level of agreement, ranging from 1 = strongly disagree to 5 = strongly agree).

We surveyed front-line miners in the Zhaojin Gold Mine in China using paper and online questionnaires. The random sampling method was used to select subjects from the front-line miners. Finally, a total of 295 valid data were obtained and the valid recovery rate was 88.6%.

#### 2.1.2. Formal Questionnaire and Sample

After the modification and processing of the pre-questionnaire, a formal questionnaire was produced. To reasonably and effectively explore the influencing factors, the second survey based on formal questions was conducted using paper and online questionnaires. A total of 362 valid questionnaires were obtained in Fankou Lead Zinc Mine in China, and the valid recovery rate was 87.5%.

### 2.2. Data Analysis

Based on the data obtained from the questionnaires, exploratory factor analysis, Pearson correlation analysis, and confirmatory factor analysis were used to ensure the statistical power with AMOS software, using SPSS (IBM, Armonk, New York, USA). The effects of influencing factors on risk perception were tested using Pearson correlation analysis and regression analysis. The mediating effects and mechanism were analyzed using SPSS. Exploratory factor analysis was used to test the structural validity of the questionnaires, and during the analysis procedure, questionnaires needed to be amended many times to form a meaningful structural dimension of influencing factors. The principal component analysis method was adopted during exploratory factor analysis, for which the judgment criteria are: interpretation of accumulated discrepancy > 50%, MSA < 0.5, and KMO. Usually, the higher the value of KMO, the higher the correlation among variables. Confirmatory factor analysis was mainly used to confirm the reliability of the structure of the influencing factors’ questionnaire. Reliability analysis was used to examine the stability and consistency of the questionnaires, wherein the better the reliability and validity of the questionnaires, the more accurate the results. The judgment criterion is Cronbach’s alpha > 0.5. Pearson correlation analysis can measure the degree of correlation among influencing factors. If the correlation between influencing factors and risk perception are significant, the influencing factors may have a linear relationship with risk perception. Then, regression analysis can be conducted to assess these relationships deeply.

#### 2.2.1. Dimensions of the Influencing Factor


*Item analysis and Exploratory factor analysis based on pre-questionnaire*


After the fundamental processing of the data, including reverse scoring, filling missing values, etc., reliability analysis showed that the Cronbach’s alpha = 0.919 (>0.7), indicating that the questionnaire with 43 items could be used to conduct data analysis.

Item analysis was conducted to delete the items that didn’t have a significant difference, including t-test, Pearson correlation analysis, reliability analysis, commonality analysis, and factor load test. If there were more than three indexes that did not reach the standard, the item would be deleted. Thus, three items were deleted. After that, the remaining 40 items were used for reliability analysis. The Cronbach’s alpha (0.934 > 0.919) increased, indicating that the internal consistency of the questionnaire improved, and exploratory factor analysis could be performed.

Exploratory factor analysis was conducted to test the construction reliability of the scale. Based on principal component analysis, 17 items were deleted. Factor 1 had 6 items that had relatively higher factor loading (0.512–0.764), indicating that the items belonged to the same scope. According to the concept of the initial preparation of the questionnaire, factor 1 was named organizational safety atmosphere. Factor 2, named risk communication, contained 4 items (factor loading: 0.629–0.826). Factor 3, named risk attitude, included 5 items (factor loading: 0.647–0.880). Factor 4, named knowledge level, covered 4 items (factor loading: 0.639–0.764). Factor 5, named organizational trust, had 4 items (factor loading: 0.469–0.745).

Then, the correlation test was conducted on the remaining 23 items. The result showed that KMO = 0.9270 (>0.5), indicating that the internal consistency of the questionnaire was favorable (Appendix A for details).


*Item analysis and exploratory factor analysis based on formal questionnaire*


Based on the pre-questionnaire, some items were adjusted based on experts’ opinions on superior questionnaires. The formal questionnaire was designed with 29 items. After the fundamental processing of the data, reliability analysis showed that the Cronbach’s alpha = 0.812 (>0.7).

Item analysis was conducted first. According to the test scores, the high group (ranking among the top 27%) and low group (ranking among the bottom 27%) were distinguished. Then, the independent samples t-test, correlation test, reliability analysis, and commonality analysis were performed to select items. A total of 8 items were deleted. The remaining 21 items were used for reliability analysis. The Cronbach’s alpha (0.900 > 0.812) increased after deletion, indicating that the internal consistency of the questionnaire improved, and exploratory factor analysis could be performed.

After exploratory factor analysis, five items were deleted. Four factors were analyzed using variance interpretation analysis (See Table 1). The interpretation of accumulated discrepancy (63.847% > 50%) was acceptable.

Factor 1 had 4 items, which had relatively higher factor loading (0.664–0.815), indicating that 4 items belonged to a unified project. These items had a closed relationship with interpersonal trust. So, factor 1 was named organizational trust. As mentioned above, Factor 2, named organizational safety atmosphere, contained 5 items (factor loading: 0.557–0.784). Factor 3, named knowledge level, included 4 items (factor loading: 0.604–0.806). Factor 4, named risk communication, covered 3 items (factor loading: 0.612–0.782).

A total of 16 items were retained to perform the correlation test. The KMO value = 0.835 (>0.5) and the significance level of the Bartlett test = 0.000, indicating that the internal consistency of the questionnaire was favorable.


*Confirmatory factor analysis*


In order to examine the stability and reliability of the four-factor structure, confirmatory factor analysis was conducted using the structural equation model (SEM). The maximum absolute value of skewness = 1.365 (<3.00) and the maximum absolute value of kurtosis = 1.924 (<10.00) in all items, so the survey data conformed to the normal distribution. Therefore, the model could be used for structural equation modeling analysis.

In order to test reliability and validity, a four-factor model was established with organizational trust, organizational safety atmosphere, knowledge level, and risk communication as potential variables (see Figure 1). After modifying the model, the chi-square value (CMIN) = 102.344 and the significant value of *p* = 0.216 (>0.05), which showed that the model had acceptable fitness (see Table 2). As for CMIN/DF = 1.112, GFI = 0.932, AGFI = 0.901 > 0.9, RMR = 0.035 (close to 0), RMSEA = 0.026 (<0.05), and CFI = 0.992 (close to 1), the data met the basic requirements and the SEM model was improved.

#### 2.2.2. Dimensions of Risk Perception

The risk perception measurement scale was designed with 15 items from three aspects: human risk, environmental risk, and equipment risk. For the first survey, item analysis was conducted. After performing a t-test, Pearson correlation analysis, reliability analysis, commonality analysis, and factor load test, one item was deleted. The remaining 14 items were used for reliability analysis. The Cronbach’s alpha = 0.872 > 0.8, indicating that the reliability is favorable.

During the second survey, item analysis was used to check the validity of the items and ensure the consistency of the scale representation. The results of the item analysis show that all items met the criteria and the t-tests were all verified. The Cronbach’s alpha = 0.843 > 0.8, which showed that the consistency of the scale was acceptable for subsequent analysis (Appendix B for details).

## 3. Results

### 3.1. Relationship between Demographic Variables and Risk Perception

The influence of age, educational level, and working years on risk perception were analyzed using the independent samples t-test and one-way analysis. The results are as follows: The values of age (F = 1.55, *p* = 0.187 > 0.05) didn’t reach a significant level, indicating that there was no significant difference in risk perception with different ages. Working years (F = 0.151, *p* = 0.929 > 0.05) was the same. The variance of educational levels (F = 4.404, *p* = 0.002 < 0.05) reached a significant level. So, there was a significant difference in risk perception for the various educational levels among miners.

### 3.2. Correlation between Influencing Factors and Risk Perception

Pearson correlation analysis was used to explore the relationships among the four explanatory variables, as well as the relationships between the explanatory variables and the dependent variable (miners’ risk perception). As shown in Table 3, there was a significant positive correlation in statistical significance. Organizational trust, organizational safety atmosphere, knowledge level, and risk communication all had a positive relationship with miners’ risk perception.

### 3.3. Effect of Single Influencing Factors on Risk Perception

A simple linear regression analysis was performed to verify the predictive effect on miners’ risk perception, with organizational trust, organizational safety atmosphere, knowledge level, and risk communication as independent variables, and miners’ risk perception as the dependent variable.

As shown in Table 4, organizational trust, organizational safety atmosphere, knowledge level, and risk communication all had a positive predictive role in miners’ risk perception.

### 3.4. Effect of Multiple Influencing Factors on Risk Perception

Based on stepwise analysis, regression analysis was performed to explore the relationships between independent variables and dependent variables, with organizational trust, organizational safety atmosphere, knowledge level, and risk communication as independent variables, and miners’ risk perception as the dependent variable.

The results show that the significance coefficient of organizational trust and organizational safety atmosphere didn’t reach a significant level, so these two variables were excluded from the regression model. As shown in Table 5, only knowledge level and risk communication had a significant predictive power for miners’ risk perception. The described variation of knowledge level and risk communication were 29.7% and 37.2%, respectively. From the perspective of standardized regression coefficients, knowledge level and risk communication both had a significant positive effect on miners’ risk perception.

### 3.5. Mediating Effects of Influencing Factors on Risk Perception

#### 3.5.1. Preliminary Model of Effect Path Analysis

Based on the results of simple linear regression and multiple regression analyses, and according to the mediator examining method [37], risk communication and knowledge level may be viewed as intermediate variables to explore the influencing mechanisms between the four influencing factors and miners’ risk perception. Model 1 (see Figure 2) was constructed, with knowledge level and risk communication as intermediate variables and miners’ risk perception as the dependent variable.

Then, a multiple-chain mediation analysis was conducted to verify model 1 of the mediation effect path. As Table 6 shows, the path from organizational safety atmosphere to knowledge level was not significant. So, it was deleted and model 2 was constructed.

#### 3.5.2. Final Model of Effect Path Analysis

Model 2 of the mediation effect path was constructed by Amos24.0 (IBM, Armonk, New York, USA) (see Figure 3). The structural equation model analysis was used to verify the influencing mechanism of model 2 and the fitting indexes are shown in Table 7 and Table 8. The chi-square degree of the freedom ratio, approximate root mean square error, GIF, AGIF, CFI, RMR, RMSE, and other indicators all meet relevant requirements and model 2 fits well.

## 4. Discussion

Firstly, in contrast with previous studies on risk perception of the public [10,38,39], the impact of age and working years on miners’ risk perception is not obvious. With an increase in age and working years, people will gain a deeper and broader understanding of health, which may change people’s health-risk perception. However, for miners, the type and content of the work is fixed. With age and working years increased, the experience obtained at work is finite, so age and working years have little influence on miners’ risk perception. Educational level largely influences miners’ risk perception, which was also found in a study conducted in Hong Kong [17]. A higher educational level indicates greater knowledge, which may contribute a more comprehensive and scientific understanding of risk. So, it is effective to develop miners’ educational level to improve their risk perception.

Secondly, the level of knowledge has a significant impact on individual risk perception, which is consistent with Olapegba’s conclusion [15]. Another study found that, because of the different levels of nuclear power knowledge, the public’s risk perception is different from that of experts [40]. Relevant knowledge has a direct impact on miners’ ability to deal with risk at work. The richer and more comprehensive the knowledge, the more accurate the miners’ perception of risk. Effective training for workers improves their capacity for risk perception [41,42]. Thus, complete in-service safety training is necessary to help miners have a comprehensive understanding of risk management, regulations, and safety regulations for operation.

Additionally, risk communication has a significant positive impact on miners’ risk perception, which is consistent with the public risk perception of flood risks [31]. Liu et al. concluded that risk communication has an obvious influence on the public’s risk perception of SARS [37,43]. Risk communication can enhance individuals’ understanding of risks, which helps them establish an appropriate perception of risks [32]. For miners working in a relatively complex environment, effective risk communication can help to avoid the occurrence of unsafe behavior. In addition to organizational files and other kinds of documents issued, risk information can be mainly obtained through risk communication, which can strengthen miners’ capacity for risk perception. The approaches of risk communication should be improved to form a virtuous communication cycle.

Moreover, the impact of organizational trust on miners’ risk perception was intermediated by knowledge level and risk communication. By researching risk perception, Zeng and Xv et al. verified that the public’s trust of source information can affect their risk perception [44]. Others’ studies also confirmed that trust in different areas, such as technology, management, and disaster all have a positive impact on individuals’ risk perception [45,46,47,48]. On the one hand, for miners, if the source, fairness, and scientificity of risk information can be trusted during risk communication, it can be more conducive to maintaining effective risk communication. On the other hand, trust in the training system and related policies is beneficial to miners’ accumulation of risk knowledge. So, managers should take steps to facilitate miners’ organizational trust on daily basis.

Finally, a proper safety atmosphere increases employees’ safety needs [49], and more safety needs require more effective risk communication. A previous study on airport security showed that the promotion of an ideal safety atmosphere is beneficial to members’ communication in a group [50]. Therefore, it is necessary to create a favorable organizational safety atmosphere to promote risk communication, which can subsequently improve miners’ risk perception.

The model and questionnaire formed for this research have the advantages of clear structure and high reliability. The results can offer support to research related to risk perception of miners, and provide a reference of empirical analysis for further study. The risk perception model we designed has important implications for safety management in companies. First, it is necessary for companies to improve workers’ organizational trust, thereby increasing credibility. Organizational trust is the beginning of the mediation path, which has an impact on workers’ motivation to learn and maintain communication. Only by assuring the organizational trust of workers can safety at work be guaranteed. Second, establishing the importance of safety through training, appropriate rewards, and punishment systems or propaganda can optimize the organizational safety atmosphere, which has a positive impact on workers’ safety attitudes and ability to avoid risk behaviors. Third, the frequent training of safety knowledge at work is necessary. Training helps workers understand safe operation rules, treatment and response to risk events, safeguards, etc. It helps workers improve their perception of accidents and enhance their capacity for early warning. Finally, it is important to increase risk information credibility by perfecting the methods of risk communication. The risk information workers receive is mostly external, so risk communication can’t be properly judged.

This research has some limitations. The survey in this study was only conducted in two mining companies, and the establishment of the questionnaire mainly drew information from other industries. So, to some extent, the results have limitations in representing miners in general. This study only focused on individual and organizational factors, but there are some additional factors that also have an impact on individuals’ risk perception, such as social and mental factors. In addition, this study only surveyed miners to explore influencing factors of risk perception and the results may be different in other careers. Further studies should conduct a more extensive and reasonable survey to improve the content of the questionnaire. For example, the questionnaire can contain more factors and the participants can be diversified, such as with different careers or different strata. Then, more advanced methods, such as cognitive neuroscience and brain imaging techniques, can be used to study individuals’ risk perception. A previous study has proven that different kinds of risks can cause different behavior in the brain [51]. Zhang et al. have already proposed the concept of safety-related psychological phenomena [52]. It is important to study how the brain behaves when people recognize risks, and to note the differences of brain activity among people who have different levels of risk perception.

## 5. Conclusions

Our results verify the impact of four factors affecting miners’ risk perception, namely: organizational safety atmosphere, organizational trust, knowledge level, and risk communication. Knowledge level (β = 0.422) and risk communication (β = 0.301) had a significant, positive direct effect on miners’ risk perception. Organizational trust has a positive indirect impact on miners’ risk perception intermediated by knowledge level and risk communication. Organizational safety atmosphere has a positive indirect impact on miners’ risk perception intermediated by risk communication.

## Figures and Tables

**Figure 1 ijerph-19-03817-f001:**
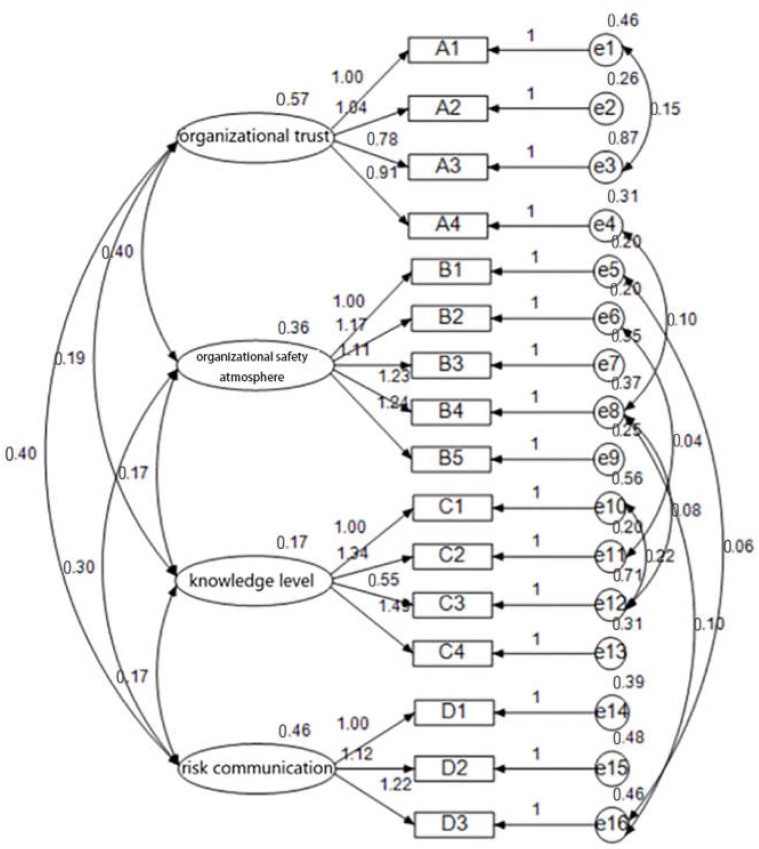
Data results of the modified model of risk perception influencing factors.

**Figure 2 ijerph-19-03817-f002:**
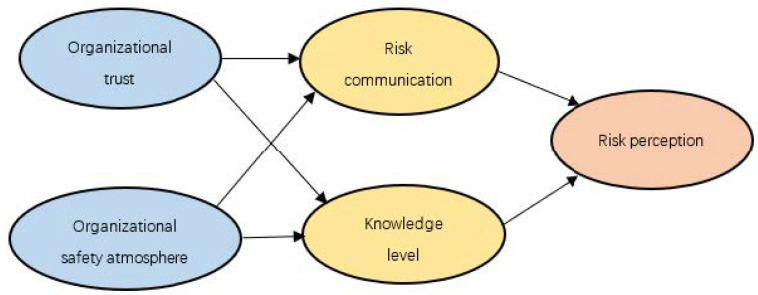
Model 1 of mediation effect path.

**Figure 3 ijerph-19-03817-f003:**
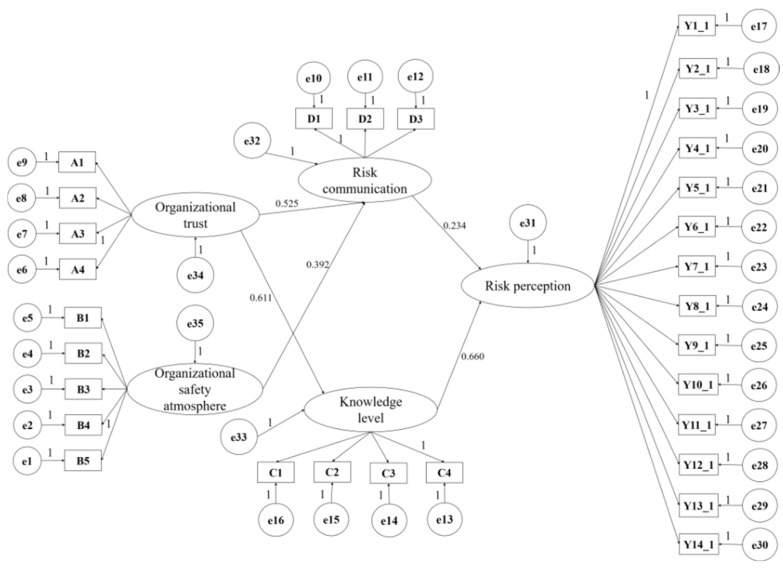
Multi-factor structural model 2 of risk perception level of miners.

**Table 1 ijerph-19-03817-t001:** Exploratory factor analysis summary of miner risk cognitive factors (second survey).

Item	Factor 1	Factor 2	Factor 3	Factor 4	New Number	Factor
X13	0.815				A1	Organizational trust
X14	0.743				A2	
X18	0.713				A3	
X12	0.644				A4	
X10		0.784			B1	Organizational safety atmosphere
X8		0.778			B2	
X7		0.669			B3	
X11		0.606			B4	
X9		0.557			B5	
X4			0.806		C1	Knowledge level
X6			0.772		C2	
X3			0.677		C3	
X5			0.604		C4	
X27				0.782	D1	Risk communication
X28				0.774	D2	
X29				0.612	D3	
KMO test	0.835					
Bartlett test	0.000					

**Table 2 ijerph-19-03817-t002:** Assumed model-fitting indicators of influential factors of miners’ risk perception.

Fitting Indicators	CMIN χ^2^	DF	CMIN χ^2^/DF	*p*	GIF	AGFI	RMR	RMSEA	CFI
Standard			<2	>0.05	>0.9	>0.9	<0.05	<0.05	>0.9
Result	102.344	92	1.112	0.216	0.932	0.901	0.035	0.026	0.992

**Table 3 ijerph-19-03817-t003:** Correlation between miners’ risk perception level and influencing factors (*n* = 362).

	Organizational Trust	Organizational Safety Atmosphere	Knowledge Level	Risk Communication	Influencing Factors	Risk Perception
Organizational trust	1					
Organizational safety atmosphere	0.677 **	1				
Knowledge level	0.337 **	0.469 **	1			
Risk communication	0.545 **	0.549 **	0.408 **	1		
Influencing factors	0.835 **	0.881 **	0.668 **	0.764 **	1	
Risk perception	0.245 **	0.363 **	0.545 **	0.474 **	0.491 **	1

** At the 0.01 level (two tailed), the correlation is significant.

**Table 4 ijerph-19-03817-t004:** Summary table of regression analysis model coefficients at various levels of influencing factors.

Independent Variable		R	R^2^	ΔR^2^	B	Betaβ	t	Significance *p*
Organizational trust	quantity	0.245	0.060	0.057	48.880		24.353	0.000
	coefficient				0.622	0.245	4.693	0.000
Organizational safety atmosphere	quantity	0.363	0.131	0.129	40.485		16.387	0.000
	coefficient				0.844	0.363	7.225	0.000
Knowledge level	quantity	0.545	0.297	0.295	28.470		11.465	0.000
	coefficient				1.787	0.545	12.063	0.000
Risk communication	quantity	0.474	0.224	0.222	38.894		19.830	0.000
	coefficient				1.624	0.474	9.986	0.000

**Table 5 ijerph-19-03817-t005:** Summary of coefficients of regression analysis model for multiple influence factors (stepwise analysis method).

Model	Test Order	R	R^2^	ΔR^2^	B	Betaβ	t	Significance *p*	Collinearity Statistics	
									Allowed Value	VIF
Intercept					22.947		9.177	0.000		
1	Knowledge level	0.545 a	0.297	0.295	1.383	0.422	9.012	0.000	0.833	1.200
2	Risk communication	0.610 b	0.372	0.369	1.034	0.301	6.444	0.000	0.833	1.200

Note: a. Predictor variable: (quantity), knowledge level; b. predictor variable: (quantity), knowledge level, risk communication.

**Table 6 ijerph-19-03817-t006:** Model 1 of path indicators.

Path	Unstandardized Coefficient	Standardized Coefficient	S.E.	C.R.	*p*
Organizational trust Risk communication	0.626	0.559	0.105	5.985	***
Organizational safety atmosphere Risk communication	0.297	0.379	0.067	4.460	***
Organizational trust Knowledge level	0.767	0.623	0.064	12.045	***
Organizational safety atmosphere Knowledge level	0.149	0.173	0.079	1.883	0.060
Risk communication Risk perception	0.138	0.234	0.036	3.862	***
Knowledge level Risk perception	0.349	0.649	0.056	6.276	***

Note: S.E.: Standard errors; C.R.: Critical values of the ratio; *p* = *** means the results are significant at 0.001 alpha level.

**Table 7 ijerph-19-03817-t007:** Fitting indicators of model 2.

Fitting Index	CMIN	DF	CMIN/DF χ^2^	*p*	GIF	AGFI	RMR	RMSEA	CFI
Standard			<2	>0.05	>0.9	>0.9	<0.05	<0.05	>0.9
Result	511.014	377	1.731	0.153	0.92	0.878	0.032	0.047	0.924

**Table 8 ijerph-19-03817-t008:** Path indicators of model 2.

Path	Unstandardized Coefficient	Standardized Coefficient	S.E.	C.R.	*p*
Organizational trust Risk communication	0.587	0.525	0.103	5.684	***
Organizational safety atmosphere Risk communication	0.309	0.392	0.068	4.575	***
Organizational trust Knowledge level	0.778	0.611	0.059	13.102	***
Risk communication Risk perception	0.140	0.234	0.036	3.888	***
Knowledge level Risk perception	0.348	0.660	0.056	6.251	***

Note: S.E.: Standard errors; C.R.: Critical values of the ratio; *p* = *** means the results are significant at 0.001 alpha level.

## Data Availability

Not applicable.

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
