# Peer review of "What Influences Miners’ Safety Risk Perception?"

_ijerph, 2022, doi:10.3390/ijerph19073817_

Round 1

Reviewer 1 Report

The manuscript presents an interesting topic and is relevant for the OHS field. The structure of the manuscript is clear and contains necessary sections. The visualization of the results is good. However, I do hope that some revisions are made. Please see my comments and questions below.

In Introduction:

  • Row 35. Is there some statistics to back up the statement that the risk of mining industry is ahead of various industries in China? Reference?
  • Row 37-38. Before the article says that unsafe behaviors are crucial causes for accidents, could you explain accidents and their causes more generally (shortly)? That way the reader will not accidentally understand that employees are blamed.
  • Row 39. When the level of risk perception was relatively low? In what situations?
  • Rows 64-69. You could explain the meaning of safety atmosphere more and what does it entail before saying what it is in this current study.
  • Row 81-83. In the article it is stated that the results are beneficial to improve miners’ risk perceptions. Could you explain more, how the results can be used to improve miners' risk perception? If not here, then in the Discussion section.

In Materials and Methods:

  • Row 87. If you refer to a verified scale, is there any other references you could use in addition to the thesis references? Anything available for reader, maybe in English?
  • Row 88. The article refers to experts, teachers and graduates who revised the initial scale. How many were there? Could you give the n -values?
  • Was the formal questionnaire also distributed by paper and network questionnaires? To how many did you send the questionnaire? Is response rate the same as valid recovery?
  • In section 2 Data analysis, it would be beneficial if you introduce the reader to the methods. Why did you choose these analyses? Why are they suitable? When is for example the value of Cronbach's alpha good? The reader may not know.

In Discussion:

  • The results could be compared little more to earlier research. Are they supporting the earlier literature or presenting something new? The discussion goes through the relevant results but compares them to very few references.
  • In rows 289-291, there is some limitations presented. However, they could be presented more broadly (for example, limitations regarding data collection, analyses, or interpretation of results). What about the possibility to repeat the study in other countries by other authors? Would they receive similar results?
  • Further studies are presented quite shortly. Could they be describer more widely? What are the most interesting aspects you would like to study further? Was there something surprising that needs more attention? What does the “more advanced methods” mean in row 293?

In Conclusions:

  • Row 304. The article ends with a sentence “Our findings provide a risk perception model.” What are your suggestions for example for companies? What should they do with the risk perception model? Or should it be studied more before use?

Language:

  • Some misspelling which distracts fluent/smooth reading.
  • For example in row 39, if you refer to two studies, then say "The studies show..." instead of singular.
  • Row 75. Maybe add “and” after the comma and before “sorted out”, that way the sentence is more clear.
  • Spaces missing for example in row 93.
  • In subtitle 1.1. Pre-questionnaire and sample. the sample is with lower case letter with a dot at the end, while in subtitle 2.1.2. Formal questionnaire and Sample, sample is in upper-case letter. None of the other titles or subtitles have a dot at the end.
  • In rows 293-296, the sentences begin with lower case letters.

Author Response

We would like to thank the editor and the reviewers for comments and their kind suggestions of our manuscript entitled “What Influence Mine Miners’ Safety Risk Perception?” (ID: ijerph-1607625). The manuscript has been revised according to their comments and suggestions. We provide this cover letter to explain, point by point, the details of our revisions in the manuscript and our responses to the reviewers’ comments as follows. In order to make the changes easily viewable for you and the reviewers, in the revised paper, we marked the revision with red color. Besides, we have carefully checked through the whole manuscript and corrected some grammar mistakes.

Reviewer 2 Report

With this paper, a questionnaire survey method was used to collect the data of risk perception from nearly 400 respondents working for the metal mine in China. Results verified four factors affecting miners’ risk perception: organizational safety atmosphere, organizational trust, knowledge level, and risk communication.

I only have the following minor comments with which to improve the paper.

Introduction section: (i) I recommend clearly defining “risk perception”; (ii) I also recommend that sections 1.1 and 1.2 have a more explicit connection with factors affecting miners’ risk perception; (iii) The last paragraph of section 1.2 (lines 74-83) could have its own section 1.3.

Materials and Methods section: (i) Please describe the sampling technique used; (ii) Have ethical aspects been considered? (iii) Why have they not been considered moderating effects?

Discussion section: (i) The research questions shown in lines 250-252 could be considered in the introduction section; (ii) Line 253 on health risks is a bit confusing since this paper is about safety risks. Therefore, please properly distinguish between safety risks and health risks; (iii) I recommend expanding the discussion of results with an additional review of the literature that allows improving comparing the results related to the four factors affecting miners' risk perception.

Author Response

(The authors gave the same response as above.)

Round 2

Reviewer 1 Report

Thank you for answering to my concerns. The revisions are well done. I suggest that the article can be published after minor revisions.

Here are the few minor notions I wish you to address:

  1. Addition in rows 39-40. Thank you for this. However, "terrible" does not seem as very scientific term. Perhaps you could say something more concrete about the working conditions? Related to language, you should not start a sentence with "and" (I noticed that in some other sentences too).
  2. In rows 345-346, what dou you mean with this sentence, "Second, strengthening importance of safety by training, appropriate rewards and punishment system or propaganda."? Is there a verb missing? Could you write this more clearly?

Again, thank you for making the revisions. Cognitive neuroscience and brain imaging technique sound interesting!

Author Response

    We would like to thank the editor and the reviewers again for comments and their kind suggestions of our manuscript entitled “What Influence Mine Miners’ Safety Risk Perception?” (ID: ijerph-1607625). The manuscript has been revised according to their comments and suggestions. We provide this cover letter to explain, point by point, the details of our revisions in the manuscript and our responses to the reviewers’ comments as follows. In order to make the changes easily viewable for you and the reviewers, in the revised paper, we marked the revision with red color. It is very kind for you to spend time reading our manuscript.

Point 1: Addition in rows 39-40. Thank you for this. However, "terrible" does not seem as very scientific term. Perhaps you could say something more concrete about the working conditions? Related to language, you should not start a sentence with "and" (I noticed that in some other sentences too).

Response 1: Considering your advice, it was an inappropriate use of words, we have corrected our words and complemented the specific situations of miners’ working conditions in the revised version (row 39-41). The language of this paper did exist some mistakes, and we have corrected the use of “And” through all this paper.

Point 2: In rows 345-346, what do you mean with this sentence, "Second, strengthening importance of safety by training, appropriate rewards and punishment system or propaganda."? Is there a verb missing? Could you write this more clearly?

Response 2: Thank you for your correction to us. This mistake that we shouldn’t make has already been revised in the revised version (row 346-349)
